# Health Risk Assessments of Selected Trace Elements and Factors Associated with Their Levels in Human Breast Milk from Pretoria, South Africa

**DOI:** 10.3390/ijerph18189754

**Published:** 2021-09-16

**Authors:** Joshua O. Olowoyo, Linda R. Macheka, Phiona M. Mametja

**Affiliations:** Department of Biology, Sefako Makgatho Health Sciences University, P.O Box 139, Pretoria 0204, South Africa; misslindamacheka@gmail.com (L.R.M.); phiona.phiolocity@gmail.com (P.M.M.)

**Keywords:** chromium, manganese, arsenic, breast milk and passive smoking

## Abstract

While breast milk is the recommended food for infants up to at least six months, exogenously derived compounds such as trace elements have been widely reported in human milk which may make it become toxic or a source of pollutants to the infants. Numerous short- and long-term health effects have been associated with high body—burdens of trace elements, which are amplified in infants. The current study determined the levels and possible contributing factors of six trace elements in breast milk of nursing mothers from a local hospital in Pretoria. Extraction of trace elements employed a digestion technique using perchloric and nitric acid in a ratio of 1:3, while Inductively Coupled Plasma–Membrane Spectrophotometry was used to identify and quantify their levels in breast milk. Concentrations of Cr and Mn were the highest in breast milk, with values ranging from 0.30 to 5.72 µg/L and 0.23 to 5.13 µg/L, respectively. Levels of Co, As, Pb and Cd ranged from <LOD to 0.2 µg/L, <LOD to 2.29 µg/L, 0.05 to 1.06 µg/L, and 0.004 to 0.005 µg/L, respectively. Levels of Cr, Mn and As were higher than the recommended limits from WHO (World Health Organization) in some milk samples. Dietary assessments showed minimal risk for the infants through breastfeeding at this stage; however, prolonged exposure to other sources of these toxic trace elements may pose a serious health risk for the infants. The nature of employment, infant birth weight, passive smoking and maternal diet were the significant factors noted to contribute to trace metal levels in breast milk.

## 1. Introduction

The long-term benefits of breastfeeding, both to infant and maternal health, have motivated the World Health Organization (WHO) to recommend it for newborns up until the age of at least six months [1,2]. Preliminary studies have shown that children who have been breastfed as infants have reduced incidence of infections and iron deficiency, are less disposed to obesity and show better intellectual performance [3,4]. However, exogenously derived elements, such as trace elements, have been widely reported in breast milk, thereby putting infants at risk of trace element accumulation and related health effects [5,6,7,8,9].

Human breast milk is a multiplex composition of live cells, fatty acids, proteins, vitamins, carbohydrates and essential elements [2,4,10]. While several studies have investigated and quantified the natural composition of trace elements such as zinc, iron, manganese and copper in breast milk [4,10,11,12], these same elements may cause detrimental health effects to breastfeeding infants, such as reduced thymus gland function, impaired cognitive development, fetal growth disorders, neurological disorders and increased neonatal fatalities, when they are present above permissible limits [13]. Accumulation of trace elements in breast milk is often a result of transfer from maternal reserves in the blood through the mammary glands [14,15], while maternal exposure occurs through the food chain, contaminated drinking water sources as well as through dermal contact and inhalation [16,17,18]. Breast milk can be used as a bioindicator for maternal exposure to pollutants and has been noted to be an essential pathway for trace element depuration for mothers [19,20].

A number of trace elements have been classified as potentially toxic and infant exposure to elements such as arsenic, cadmium and lead has been associated with anaemia, various cancers, interference of bone development leading to rickets, negative effects on the nervous system, as well as autism [20,21]. Maternal arsenic exposure through contaminated water in some parts of the world, such as Bangladesh and Iran, have been reported to lead to increased infant death. It has also been noted that inorganic forms of As and Pb are often stored in maternal blood and bone, respectively, through sequestration [22,23]. In addition, previously published provisional tolerable daily intakes for arsenic (2.1 µg/kg-BW/day) were considered unsafe for humans, and in 2010, the Joint FAO/WHO Expert Committee on Food Additives (JECFA) increased this limit to 3 µg/kg-BW/day. Levels of Pb in breast milk reflect both endogenous and external exposure to the element. Published data have shown that pregnancy and breastfeeding intensify bone turnover, and therefore increase mobilization of Pb from the maternal skeleton [24,25]. The release of Pb from the skeleton, however, is partially regulated by Ca intake, where low dietary calcium poses as a risk factor for increased sequestrated Pb levels [24]. While blood is the acceptable biomarker of cadmium levels in humans [17,26], there are several studies that have reported on its occurrence in human breast milk [27,28,29,30,31].

Although arsenic, lead and cadmium are some of the most studied trace elements in human breast milk, elevated levels of cobalt, chromium, mercury and manganese have also been detected in breast milk [32,33,34,35]. Cobalt, chromium and manganese are considered to be micro-elements needed in small amounts for the proper growth and function of the human body [36,37,38]. Apart from being carcinogenic, high exposure to these trace elements may lead to organ toxicity and delayed hypersensitivity [39]. Cobalt exposure has been noted to be through contaminated soil and water, and elevated levels in the human body have been linked to the development of goiter [40]. Although cadmium has no known physiological function in the human body [41], high levels of this element have been considered to be carcinogenic as the element stimulates cell proliferation [42]. In addition, high Cd levels have also been associated with menstrual cycle disorders, tissue damage in the liver and kidney and alteration in reproductive hormones [41]. The accumulation of some trace elements, among other contaminants in breast milk, like phthalates and organo-chlorine pesticides, correlated to its high protein and fat content [43]. This is due to the high protein or fatty acid affinity of elements such as zinc or cadmium, which can occupy specific binding sites on these organic molecules [44,45].

The scarcity of data from the African continent on infant exposure to a number of trace elements through breastfeeding creates a knowledge gap with regards to risk assessments. Several studies conducted in Pretoria, South Africa, previously on the levels of trace elements in water, vegetables, soil and human samples (blood and urine) suggested elevated levels of toxic trace elements [46,47]. The study by Macheka et al. (2016) [48] on trace elements in pregnant women practicing geophagia showed that toxic trace elements in their bloods were higher than the recommended limits, with levels of Cr (5.78 ± 4.40 µg/L), Pb (2.90.78 ± 1.40 µg/L) and Mn (22.42 ± 14.10 µg/L) reported to be above the acceptable values of 0.23 µg/L, 0.80 µg/L and 12.6 µg/L, respectively. It is therefore prudent to check if these toxic trace elements are present in the breast milk that is given to the infants. This study, therefore, investigated the levels of three micro-elements: cobalt (Co), chromium (Cr) and manganese (Mn); and three potentially toxic elements: arsenic (As), lead (Pb) and cadmium (Cd) in breast milk from a group of South African women, and further examined factors associated with the accumulation of these trace elements in their breast milk.

## 2. Materials and Methods

### 2.1. Study Population and Sample Collection

Prior to sampling, the study protocol, which described in detail the cross-sectional and analytical nature of the study, was reviewed and approved by an Ethics Committee from Sefako Makgatho Health Sciences University, Pretoria, South Africa (Ref: SMUREC/S/29/2018:PG). Following informed consent, participants were recruited from a postnatal ward at a local hospital in Pretoria, Gauteng Province, South Africa. Fifty-four lactating women took part in the study, who first filled out a structured questionnaire collecting data such as sociodemographic information, maternal and infant health status, maternal diet and drinking water sources, among others. Breast milk samples were manually expressed in polypropylene tubes which were pre-cleaned with methanol and milli-Q water (conductivity 18 M Ω cm^−1^ (18 S cm^−1^)); and kept at −20 °C before analysis. Samples were collected between February and October 2020, and all COVID-19 safety protocols were observed during sampling. Only exclusively breastfeeding women who had given birth within a month were included in the study, and these women were assigned a study code to maintain anonymity during the study.

### 2.2. Sample Analysis

Extraction of trace metals employed a digestion technique using perchloric and nitric acid in a 1:3 ratio. In this technique, 3 mL of perchloric acid and 9 mL of 65% concentrated nitric acid were added to 1 g of each sample in a quartz tube and the mixture vortexed for 30 s. The acidified milk samples were then digested at 90 °C in an oven for two hours, then cooled to room temperature. Following this, the acidified samples were dissolved in 5 mL of 1.5% nitric acid, and this was made up to a 25 mL volume using milli-Q water (conductivity 18 M Ω cm^−1^(18 S cm^−1^)). The resultant solution was then filtered using Whatman filter paper No. 42, and subsequently, 70 µL of Internal Standard (IS) corresponding to each element of interest were added to each sample.

Identification and quantification of trace elements in the breast milk samples were carried out at an accredited laboratory by the use of an Agilent 7700 series Inductively Coupled Plasma–Membrane Spectrophotometer (ICP–MS) following standard procedure. Three solvent blanks were included in the run. Certified reference material was used to calibrate the instrument before introducing the blanks, and once a line of best fit was obtained for each element of interest, the samples were then injected into the ICP–MS. Recalibration was also done after every 10 samples. To improve precision and reduce analytical bias, samples were analyzed in triplicate and both internal quality control standards and certified reference materials specific to each trace metal were used to verify and validate the results. The limits of detection (LOD), limits of quantitation (LOQ) and the percentage expanded uncertainty (%U_exp_) are presented in Table 1.

### 2.3. Statistical Analysis

For statistical purposes, all concentrations which were quantified below the limit of detection (<LOD) were calculated as LOD/2. Microsoft Excel^®^ 2019 was used for all the descriptive statistics, while IBM SPSS software (version 26; Chicago, IL, USA) was used to perform nonparametric (Kruskal–Wallis and Spearman’s Correlation) tests. The statistical significance was set at *p* < 0.05 (one-tailed).

### 2.4. Dietary Risk Assessments

Dietary exposure to Co, Cr, Mn, As, Pb and Cd through breastfeeding was estimated using the USEPA guidelines [49], which state that an exclusively breastfed infant would weigh an average of 4.6 kg within their first month and would consume approximately 510 mL/day of breast milk. The upper percentile of breast milk consumption was set at 950 mL/day according to the EPA guidelines. The estimated daily intake of the trace elements was calculated as follows:(1)$$\mathrm{EDI}\;({\mathrm{ngkg}}^{-1}{\mathrm{BWday}}^{-1})=\frac{\mathrm{CTrace}\;\mathrm{Metals}\;\times \;\mathrm{DVBreastmilk}}{\mathrm{BW}}$$
where EDI (µg/L^−1^BWday^−1^) is the estimated daily intake of trace elements through breastfeeding, C_Trace Elements_ is the mean concentration (µg/L) of each trace metal in the breast milk, DV_Breastmilk_ is the consumed daily volume of breast milk (mL/day), and BW (expressed in kg) is the average body weight of the infant according to EPA guidelines.

The Hazard Quotients (HQ) for the trace elements were calculated to determine the noncarcinogenic risk according to EPA guidelines [50]. The noncarcinogenic risk was determined as follows:(2)$$\mathrm{HQ}=\frac{\mathrm{EDI}\;\left(µg/L/\mathrm{BW}/\mathrm{day}\right)}{\mathrm{RfD}}$$
where HQ is the Hazard Quotient, EDI is the estimated daily intake of trace elements through breast milk consumption and RfD is the Reference Dose, according to the World Health Organization Guideline Values [51]. If HQ < 1, the risk on the infant will be considered as negligible, and if HQ > 1, the risk will be considered as an unacceptable risk posing a noncarcinogenic health threat to the infant [50].

## 3. Results and Discussion

### 3.1. Trace Metal Levels in Breast Milk

The descriptive statistics of the concentrations of the trace elements in breast milk are presented in Table 2, together with comparisons with the acceptable ranges of these elements specifically in human breast milk [52] and general dietary guideline values according to the World Health Organization [51]. The results showed the distribution levels of trace elements in breast milk to be in the order Cr > Mn > As > Pb > Co > Cd (Table 2).

Trace elements were detected above the LOD in at least 71% of all the samples, with Co being the least detected element. The highest median levels of 0.501 µg/L and 0.488 µg/L were from Cr and Mn, respectively. From some breast milk samples, the highest values obtained for these two elements were above the acceptable values and above the published guideline values set by the WHO [52].

The median levels of Cr in the current study were higher than those described in the USA [53] and Finland [54], who reported mean levels of 0.18 ng/mL and 0.4 ng/mL, respectively. However, Cr levels in breast milk reported from Iran were much higher than in the present study, with median levels of 3.95 µg/L, and these levels were attributed to exposure through contaminated foodstuff [13]. The median Mn levels from breast milk were significantly lower (*p* < 0.05) than those reported in literature, with most other studies reporting values ≥ 1 µg/L [35,55,56,57,58]. A study by Mitchell et al. (2020) summarized the levels of Mn in breast milk reported in more than 20 different countries from January 1980 to December 2017. From this summary, levels of Mn in 5 423 breast milk samples ranged from 0.17 to 30.27 µg/L, and a mean of means of 7.7 µg/L was established [35].

Co levels in this study could only be compared to the recommended nutrient intake in its organic form of Vitamin B_12_ as there is currently no WHO recommended dietary allowance set for this as an inorganic element. In its organic form, Co plays a role in the synthesis of various amino acids and neurotransmitters, but has been associated with fibrosis and the overproduction of erythrocytes [61]. The Co levels in the current study were within similar ranges of those reported in other parts of the world such as Poland, the Czech Republic, Germany, the United Arab Emirates and Sweden [32,58]. It has been noted that Co levels in breast milk tend to increase with the increase in lactation period, so as to cater to the growing demands of the infant as they will be producing increased humoral antibodies over time [62]. This could possibly explain the very low levels of Co in this study, as samples were collected within one week of delivery, when the demand of this element is not yet essential.

In comparison, the levels of As, Pb and Cd in the breast milk were lower than those of Co, Cr and Mn, with median levels of 0.116 µg/L, 0.090 µg/L and 0.009 µg/L for As, Pb and Cd, respectively. The levels of Pb and As in the present study were lower than those reported in Argentina, Namibia, Poland and the USA [12] and Jordan [63] However, the mean concentrations of As, Pb and Cd in this study were within similar ranges to those found in the United Arab Emirates, where levels of 0.089 ± 0.078 µg/L, 0.019 ± 0.055 µg/L and 0.003 ± 0.008 µg/L, respectively, were reported [33]. The highest concentrations of As in this study were higher than the acceptable limit set by WHO [52]. Increased concentration of As in infants may increase the risk of cancer at a later stage [64].

Among the potentially toxic elements in this study, As has the shortest half-life of 4–10 days [23,65], and the low concentrations found in breast milk could be indicative of past or less recent exposure. On the other hand, the long half-lives of Pb and Cd, which range between 10 and more than 30 years [41,66], may be reflective of the slow excretion rate of these elements through breast milk.

### 3.2. Estimation of Daily Intake and Risk Assessment

The daily intake (EDI) and hazard quotients (HQ) of trace elements through breast milk consumption were estimated (Table 3). The EDI was low for both average and upper percentile consumers, owing to the low levels of trace elements in the samples collected (Table 3). The HQ for Co, Cr, Mn, As, Pb and Co were all <1, indicating negligible risk to the nursing infants. The potential health risks posed by trace metal accumulation through breastfeeding in this study were minimal, compared to preliminary studies from other parts of the world [18,57,67,68,69]. In Iran, Samiee et al. [13] report significant risk (HQ > 1) for Pb and As in breast milk, while India [67] and Portugal [57] reported EDIs of 2.57 µg/kg-BW/day and 0.82 µg/kg-BW/day for As, both above the WHO acceptable ranges in breast milk [59].

### 3.3. Correlations between Trace Elements in Breast Milk

Table 4 shows the correlations of trace elements in breast milk with each other. Co, Cd, Mn and Cr showed positive correlations with all the other elements analyzed in this study, with significance ranging from *p* = 0.000 to *p* = 0.027. While Pb showed strong correlations (*p* < 0.05) with all the other elements, its relationship with Mn was not significant (*p* = 0.114). The correlation between essential and potentially toxic trace elements in breast milk is probably a reflection of their similar physiological pathways [23]. This has been noted especially for elements such as calcium and lead, which both end up in the bone tissue through the same uptake routes [23,70]. This same phenomenon could explain the strong correlations among most trace elements in this study.

### 3.4. Possible Factors Influencing Trace Metal Levels in Breast Milk

Table 5 summarises the sociodemographic data of the participants and trace element levels for each group, while Table 6 shows the characteristics of infants in relation to the trace element concentrations of the breast milk. In brief, the participants’ ages ranged from 15 to 38 years, with 43% of them being primiparous, while 57% had 2–6 children. None of the participants were smokers, although 13% of them were exposed to cigarette smoke through a partner or family member who was a smoker (Table 5). While none of the infants had congenital malformations, 31% of them were born prematurely (Table 6).

Although some studies have reported factors such as maternal age, parity and infant sex to influence the levels of trace elements in breast milk [23,71], the current study found no significant relationship of trace metal levels with maternal age, maternal weight, maternal education, parity, infant sex, feeding frequency and sources of drinking water (Appendix A). These observations, however, are similar to those reported in many other studies in literature [17,20,58,72,73].

There is conflicting evidence on the influence of maternal age on trace metal levels in breast milk. Where some studies, like the present one, have found no substantial relationship between the two, other studies have reported significantly higher levels of trace elements in older women, such as in Saudi Arabia [74], Japan [75], India [76], Slovakia [77] and Taiwan [71]. On the other hand, studies from industrial Zarinshahr, Iran [78] show that trace metal levels were significantly higher in younger women, and in Japan [79] and Hamadan, Iran [9], although it was noted that younger women had higher levels of trace elements, the difference was not significant.

Parity did not have any significant correlation (*p* > 0.05) to any of the trace metal levels, and this observation was similar to the study in Argentina, Namibia, Poland and USA [12]. There were significant correlations noted between Cd levels and employment status (*p* = 0.006), suggesting possible occupational exposure among the South African women. Although the women who were either employed or self-employed reported not to be occupationally exposed to any trace metal, levels of Cd in their breast milk were slightly elevated compared to unemployed women (Table 5). This was contrary to most studies where no substantial relationships between employment status and trace elements were noted [9,13,20,73]. However, Cd was found significantly higher in the breast milk of unemployed women in a few studies [14,80], and strongly correlated to working conditions in a study in Iraq [81].

While the current study found no significant relationship between concentrations of trace elements in breast milk with infant sex, Pb levels have been documented to occur at significantly higher levels in infant males than females [72,82]. This trend has been observed to continue throughout the life of the individual, whereas Cd levels have been reported to be higher in adult females than males, indicating that sex has some role in trace metal accumulation [17,82]. Although there was no statistical difference in the PB levels in breast milk of mothers of male and female infants, it was observed that Pb concentrations were indeed slightly higher in breast milk of mothers of males than mothers of female infants (Table 6). The current study did not determine the differences in the concentration of trace metals in the blood of the infants at this stage, and future studies should develop a mechanism that will allow them to follow through.

Co levels in South African breast milk were found to positively correlate with infant birth weight (*p* = 0.013) (Appendix A). There are currently very few reports that have studied Co in human breast milk, and none of them have reported on possible factors influencing its levels, and thus no comparisons with literature data could be made. The current study also found a significant positive relationship (*p* = 0.006) between Mn levels and passive smoking among the mothers (Appendix A), which have also not been reported elsewhere. Other trace metals, however, such as cadmium and lead, have been shown to increase in breast milk in women exposed to cigarette smoke during pregnancy [31].

Among the potential dietary factors, the consumption of tinned fish (*p* = 0.035), seafood (*p* = 0.002) and dairy milk (*p* = 0.041) showed significantly positive relationships with levels of Cr (Appendix A). The diet has been one of the major exposure routes to trace elements, and seafood in particular has been a major source of potentially toxic elements like As and Hg, owing to the biomagnification of these elements in aquatic environments [23,28,83]. This was in line with the observations in the current study, where positive correlations of As (*p* = 0.033), Pb (*p* = 0.017), Cd (*p* = 0.049) and Mn (*p* = 0.014) were noted among seafood consumers (Appendix A). As has been the most frequently documented potentially toxic element in seafood, with inorganic As being reported in shellfish, seaweed and algae [18,67,84]. Consumption of seafood has also been linked to maternal exposure to Cd, which could later be depurated through breast milk [28,83].

The major limitation of this study was the restricted number of samples, which could reduce the precision of our results. Trace metal analysis in clinical samples is in itself a difficult task, as there are numerous factors that play a role in the occurrence of these elements in the human body. A larger sample size on a larger scale could have given more insight into the factors associated with the levels of trace elements in the breast milk. Secondly, the study population was recruited from a single hospital, and may not be a true reflection of the entire South African population, with regards to the study at hand.

## 4. Conclusions

Levels of most of the elements were found to be below the WHO dietary guideline values and within the acceptable ranges in breast milk. However, there are instances where the levels were above the permissible limit in breast milk, and this may pose an additional health risk for the infants, especially if living in a polluted environment. This is the first documented study of trace metal levels in breast milk from South Africa and contributes to global data showing a decline in trace metal levels in breast milk over the years. Dietary risk assessments through estimated daily intakes and hazard quotients showed that the consumption of breast milk from the present study population posed negligible health threat to the infants, for both average consumption and upper percentile consumption.

While the study found no significant relationships between trace metal levels in breast milk and maternal age, maternal weight, maternal education, parity, infant sex, frequency of breastfeeding and drinking water sources, to the best of our knowledge, it is the first to find significant correlations between Co levels in breast milk and infant birth weight, as well as Mn levels and passive smoking among nursing mothers. Tinned fish, seafood and dairy milk consumption were strongly correlated with potentially toxic elements in the study, and further investigation into these dietary sources is required. In addition, there has been evidence that transfer of trace elements from both endogenous and exogenous sources into breast milk is very slow and could explain the low levels of trace elements in breast milk, compared to actual maternal exposure. In this regard, further study comparing matching maternal serum samples to breast milk samples, on a larger scale, could give more insight into the levels of trace metal exposure for both the mother and their nursing infants.

## Figures and Tables

**Table 1 ijerph-18-09754-t001:** Limits of detection (LOD), limits of quantitation (LOQ) and percentage expanded uncertainty (%Uexp) of trace elements in breast milk (µg/L).

Parameters	Co	Cr	Mn	As	Pb	Cd
LOD	0.018	0.079	0.003	0.068	0.005	0.004
LOQ	0.061	0.264	0.011	0.226	0.016	0.015
%U_exp_	2.584	2.792	3.386	1.870	1.331	1.904

**Table 2 ijerph-18-09754-t002:** Descriptive statistics of trace elements in human breast milk (n = 56) from Pretoria, compared to acceptable ranges in breast milk and guideline values by the World Health Organization (µg/L).

Trace Elements	Detection Frequency	Min	Max	Median	Mean	SD	WHO Acceptable Ranges in Breast Milk ^a^	WHO Dietary Guideline Values ^b^
Co	71%	<LOD	0.274	0.023	0.036	0.044	0.15–0.35	0.40 ^c^
Cr	100%	0.300	5.722	0.501	0.689	0.773	0.80–1.50	50
Mn	100%	0.236	5.131	0.488	0.664	0.729	3.00–4.00	40
As	89%	<LOD	2.298	0.116	0.166	0.304	0.20–0.60	10
Pb	100%	0.054	1.056	0.090	0.125	0.143	2.00–5.00	10
Cd	100%	0.004	0.053	0.009	0.013	0.011	<1	3

^a^ Acceptable ranges of trace elements in human breast milk [59]. ^b^ Guideline values based on the World Health Organization [51]. ^c^ Recommended nutrient intake in the form of Vitamin B_12_ [60].

**Table 3 ijerph-18-09754-t003:** Estimated Dietary Intake (EDI) (µg/L-BW/day) and Hazard Quotients (HQ) of trace elements through breast milk consumption.

Trace Elements	Average Consumption		Upper Percentile Consumption	
	EDI	HQ	EDI	HQ
Co	0.004	0.010	0.008	0.020
Cr	0.076	0.002	0.142	0.003
Mn	0.074	0.002	0.137	0.003
As	0.018	0.002	0.034	0.003
Pb	0.014	0.001	0.026	0.003
Cd	0.001	0.000	0.002	0.001
Mean EDI Σ_6_ Trace Elements	0.187		0.349	

**Table 4 ijerph-18-09754-t004:** Correlations of trace elements in breast milk from South Africa.

		Co	Cr	As	Pb	Cd	Mn
Co	Correlation Coefficient	1	0.300	0.529	0.337	0.481	0.689
	Significance		0.027	0.000	0.013	0.000	0.000
Cr	Correlation Coefficient	0.300	1	0.308	0.622	0.432	0.400
	Significance	0.027		0.023	0.000	0.001	0.003
As	Correlation Coefficient	0.529	0.308	1	0.218	0.418	0.571
	Significance	0.000	0.023		0.114	0.002	0.002
Pb	Correlation Coefficient	0.337	0.622	0.218	1	0.588	0.411
	Significance	0.013	0.000	0.114		0.000	0.002
Cd	Correlation Coefficient	0.481	0.432	0.418	0.588	1	0.551
	Significance (1-tailed)	0.000	0.001	0.002	0.000		0.000
Mn	Correlation Coefficient	0.619	0.400	0.517	0.411	0.551	1
	Significance (1-tailed)	0.000	0.003	0.000	0.002	0.000	

**Table 5 ijerph-18-09754-t005:** Sociodemographic characteristics of the study population (n = 54) and mean (SD) trace element levels in breast milk (µg/L).

Maternal Characteristics		N (%)	Co	Cr	As	Pb	Cd	Mn
Maternal Age (years):		54						
Maternal Weight (kg)		54						
Marital Status:	Single	35 (65)	0.037 (0.045)	0.632 (0.300)	0.130 (0.082)	0.110 (0.065)	0.014 (0.011)	0.695 (0.838)
	Married	12 (22)	0.037 (0.051)	0.885 (1.529)	0.290 (0.633)	0.163 (0.282)	0.011 (0.013)	0.616 (0.597)
	Living with Partner	7 (13)	0.035 (0.021)	0.687 (0.694)	0.140 (0.068)	0.142 (0.091)	0.010 (0.004)	0.142 (0.378)
Parity	Primiparous	23 (43)	0.048 (0.063)	0.918 (1.138)	0.221 (0.456)	0.165 (0.204)	0.015 (0.012)	0.719 (0.525)
	Multiparous	31 (57)	0.028 (0.018)	0.530 (0.236)	0.126 (0.083)	0.098 (0.064)	0.012 (0.010)	0.626 (0.867)
Maternal Education:	Primary	3 (6)	0.037 (0.022)	0.709 (0.466)	0.113 (0.028)	0.117 (0.024)	0.011 (0.003)	0.716 (0.520)
	Secondary	36 (67)	0.041 (0.052)	0.777 (0.934)	0.191 (0.370)	0.137 (0.173)	0.014 (0.012)	0.744 (0.879)
	Tertiary	15 (27)	0.027 (0.015)	0.497 (0.160)	0.119 (0.051)	0.103 (0.050)	0.011 (0.010)	0.469 (0.153)
Maternal Employment	Unemployed	46 (85)	0.039 (0.047)	0.720 (0.835)	0.177 (0.328)	0.127 (0.153)	0.008 (0.010)	0.703 (0.790)
Status:	Self-Employed	2 (4)	0.045 (0.012)	0.708 (0.448)	0.114 (0.005)	0.213 (0.136)	0.010 (0.013)	0.437 (0.174)
	Employed	6 (11)	0.019 (0.013)	0.505 (0.227)	0.106 (0.060)	0.089 (0.023)	0.013 (0.002)	0.459 (0.201)
Occupational Exposure	Yes	0						
to Chemicals/Trace Elements:	No	8 (100)						
Monthly Household	<R5000	43 (80)	0.040 (0.048)	0.700 (0.819)	0.179 (0.339)	0.129 (0.157)	0.014 (0.011)	0.702 (0.815)
Income (Rands):	R5000—R10,000	7 (17)	0.036 (0.023)	1.362 (1.258)	0.180 (0.134)	0.190 (0.112)	0.012 (0.006)	0.720 (0.294)
	>R10,000	2 (4)	0.021 (0.009)	0.524 (0.391)	0.106 (0.056)	0.099 (0.060)	0.010 (0.009)	0.478 (0.205)
Smoking During	None	47 (87)	0.037 (0.046)	0.710 (0.828)	0.170 (0.326)	0.133 (0.153)	0.014 (0.012)	0.571 (0.405)
Pregnancy	Passive Smoker	7 (13)	0.034 (0.021)	0.596 (0.236)	0.143 (0.065)	0.079 (0.016)	0.010 (0.003)	0.653 (1.728)
	Smoker	0						

**Table 6 ijerph-18-09754-t006:** Characteristics of infants in the study population (n = 54) and mean (SD) trace element levels in breast milk (µg/L).

Infant Characteristics Title		N (%)	Co	Cr	As	Pb	Cd	Mn
Infant Sex:	Male	36 (67)	0.043 (0.051)	0.765 (0.930)	0.187 (0.367)	0.138 (0.168)	0.015 (0.012)	0.784 (0.871)
	Female	18 (33)	0.024 (0.165)	0.557 (0.302)	0.127 (0.097)	0.104 (0.072)	0.010 (0.007)	0.429 (0.190)
Infant Gestation Period:	Full Term	37 (69)	0.040 (0.051)	0.696 (0.884)	0.185 (0.360)	0.127 (0.166)	0.012 (0.010)	0.743 (0.860)
	Premature	17 (31)	0.030 (0.018)	0.695 (0.499)	0.127 (0.110)	0.125 (0.082)	0.015 (0.013)	0.498 (0.296)
Infant Birth Weight (kg):		54						
Birth Abnormalities	None	54						
Estimated Feeding	Every 30 min	6 (11)	0.024 (0.017)	0.582 (0.174)	0.101 (0.037)	0.082 (0.012)	0.008 (0.003)	0.496 (0.327)
Frequency	Every Hour	4 (7)	0.026 (0.017)	0.669 (0.260)	0.109 (0.070)	0.090 (0.019)	0.008 (0.003)	0.490 (0.272)
	Every 2 Hours	16 (30)	0.026 (0.018)	0.669 (0.542)	0.123 (0.073)	0.115 (0.077)	0.012 (0.007)	0.782 (1.179)
	Every 3 Hours	28 (52)	0.047 (0.057)	0.739 (1.000)	0.214 (0.416)	0.094 (0.190)	0.010 (0.014)	0.661 (0.494)

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
