# Peer review of "Health Risk Assessments of Selected Trace Elements and Factors Associated with Their Levels in Human Breast Milk from Pretoria, South Africa"

_ijerph, 2021, doi:10.3390/ijerph18189754_

Round 1
Reviewer 1 Report
The authors provided heavy metal biomonitoring data using breast milk, which is very important evidence to show their actual risk of exposure. I have a comment for minor revision.
The authors presented the description of socio-demographic characteristics of mothers and characteristics of infacts. Then, the authors did a statistical analysis to investigate the difference exposure by each factor. For example, "the current study found no significant (p < 0.05) relationship of trace metal levels with maternal age, maternal weight, maternal education, parity, infant sex, feeding frequency and sources of drinking water", or "There were significant correlations noted between Cd levels and employment status".
The authors just checked the correlation, but it is suggested to compare the mean levels by different categories and present them with p-value. If sample size is small, authors can consider the non-parametric method (e.g. Kruskal-Wallis test) to compare different means.
Then, the authors should include the average (SD) of the level of each metal in mothers' breast milk in Tables 5 and 6 for each category and put "star (*)" signs if there are statistical significances".
N(%) Maternal Metal Level (ug/L)
Mean (SD)
Infant Gestational Period Cd As Mn
Full 37 (69) 15.2 (2.1) 5.8 (1.2)
Premature 17 (31) 19.5 (1.1)* 5.3 (2.1)
* Category with a significantly higher level
Based on this table, we can figure out the mothers of premature infants had higher levels of Cd and how much the means were different. This is just an example. It would be more informative to modify your tables 5 and 6 like this.
Author Response
The authors would like to thank the reviewers for their valuable input and suggestions in refining this manuscript. Please find below our responses to their comments:
Reviewer 1
Comments and Suggestions for Authors
The authors provided heavy metal biomonitoring data using breast milk, which is very important evidence to show their actual risk of exposure. I have a comment for minor revision.
The authors presented the description of socio-demographic characteristics of mothers and characteristics of infacts. Then, the authors did a statistical analysis to investigate the difference exposure by each factor. For example, "the current study found no significant (p < 0.05) relationship of trace metal levels with maternal age, maternal weight, maternal education, parity, infant sex, feeding frequency and sources of drinking water", or "There were significant correlations noted between Cd levels and employment status".
The authors just checked the correlation, but it is suggested to compare the mean levels by different categories and present them with p-value. If sample size is small, authors can consider the non-parametric method (e.g. Kruskal-Wallis test) to compare different means.
Thank you for this. Supplementary Information has been added to present the data.
Then, the authors should include the average (SD) of the level of each metal in mothers' breast milk in Tables 5 and 6 for each category and put "star (*)" signs if there are statistical significances".
N(%) Maternal Metal Level (ug/L)
Mean (SD)
Infant Gestational Period Cd As Mn
Full 37 (69) 15.2 (2.1) 5.8 (1.2)
Premature 17 (31) 19.5 (1.1)* 5.3 (2.1)
* Category with a significantly higher level
Based on this table, we can figure out the mothers of premature infants had higher levels of Cd and how much the means were different. This is just an example. It would be more informative to modify your tables 5 and 6 like this.
Thank you for this. Tables 5 and 6 have been modified to present the missing data.
Reviewer 2 Report
The manuscript focuses on the assessment of the selected trace elements in the breast milk of lactating women from South Africa and estimates the potential health risks of infants through breast milk consumption. Regardless the small sample size, these are the first results on trace elements levels in the African population and are as such an important contribution to current literature. However, the manuscript has to improve in its quality in order to be accepted for publication in the journal. My general comments are below and specific comments are in the attached pdf file of the manuscript.
GENERAL COMMENTS:
- Please unify the terms when referring to the elements throughout the whole manuscript. Used were many different terms such as potentially toxic elements, trace minerals, heavy metals, trace metals (although As which is metalloids was included), etc. which is confusing for a reader. Use one term only such as trace elements or trace metal(loids). Avoid using heavy metals as there are no clear definitions of this term in the literature and lately its use has been not advised (DOI: 10.1007/s11631-021-00468-0).
- The description of trace elements analysis part lacks major details (sample preparation, method parameters) and needs to be presented in a way to enable its repetition by other researchers.
- The statistical analyses seem to be inappropriate or not clearly presented, there is a mixture of parametric and non-parametric tests and results do not support the described method of statistics.
- The big part of the manuscript is on factors associated with trace elements levels (it is also in the title of the manuscript) however the results of this statistical analysis are not presented in the article (added are only p-values in the text). In my opinion, the actual level differences between various groups with added statistical significance (e.g. employed and unemployed women and Cd levels) should be in the table.
- English of a manuscript needs to be significantly improved.

Author Response
Reviewer 2
Comments and Suggestions for Authors
The manuscript focuses on the assessment of the selected trace elements in the breast milk of lactating women from South Africa and estimates the potential health risks of infants through breast milk consumption. Regardless the small sample size, these are the first results on trace elements levels in the African population and are as such an important contribution to current literature. However, the manuscript has to improve in its quality in order to be accepted for publication in the journal. My general comments are below and specific comments are in the attached pdf file of the manuscript.
GENERAL COMMENTS:
- Please unify the terms when referring to the elements throughout the whole manuscript. Used were many different terms such as potentially toxic elements, trace minerals, heavy metals, trace metals (although As which is metalloids was included), etc. which is confusing for a reader. Use one term only such as trace elements or trace metal(loids). Avoid using heavy metals as there are no clear definitions of this term in the literature and lately its use has been not advised (DOI: 10.1007/s11631-021-00468-0).
Thank you for this, the term “trace elements” has now been used throughout the manuscript.
- The description of trace elements analysis part lacks major details (sample preparation, method parameters) and needs to be presented in a way to enable its repetition by other researchers.
Thank you, the details of the methodology have been added to the manuscript.
- The statistical analyses seem to be inappropriate or not clearly presented, there is a mixture of parametric and non-parametric tests and results do not support the described method of statistics.
Statistical analyses have been redone.
- The big part of the manuscript is on factors associated with trace elements levels (it is also in the title of the manuscript) however the results of this statistical analysis are not presented in the article (added are only p-values in the text). In my opinion, the actual level differences between various groups with added statistical significance (e.g. employed and unemployed women and Cd levels) should be in the table.
Thank you, the information on the factors has been added as supplementary material to the manuscript.
- English of a manuscript needs to be significantly improved.
English of the manuscript has been improved.
Specific comments and suggested have been responded to in the attached document.
Round 2
Reviewer 2 Report
The manuscript improved significantly and the authors considered most of the suggestions. However, after the second reading, I still have some comments and suggestions (also because I did not receive the author's response on specific comments attached in the first manuscript), therefore I suggest the article to be approved for publication after minor revision.
INTRODUCTION:
line 65 -66: However, in the case of Pb it is known that Pb from bone is being released during pregnancy and lactation (see Gulson et al.2003)- due to that there would be more Pb in breast milk which does not add up with this statement. Moreover, as already stated in the first review I am missing in this article some information on the fact that in the case of Pb there is an endogenous exposure in addition to external one during the pregnancy and lactation due to this mobilization of Pb from the skeleton. The manuscript just mentioned that Pb is stored in the bone, but the crucial information on its release from bone during pregnancy and lactation is missing and should be added (in the introduction and maybe in factors influencing Pb levels in breast milk).
line: 69-71: This sentence does not add valuable information to the introduction (the same info as in sentence in lines 77-78). Similarly as for As add information if Cd levels were high, above tolerable levels or if levels are in general low.
METHODS:
line 160: change the sign with p < 0.05
RESULTS & DISCUSSION
line 279: maybe add a range of the correlation factors
line 282: there seems to be a mistake here as there are to different p-values
line 311: no need to add p < 0.05 here as you already specified in methods what you mean by significance. Also no significant would then be p>0.05. This p < 0.05 is confusing so I suggest removing it.
line 365 - 373: This paragraph is a bit confusing. Here you are discussing the maternal levels in relation to the sex of their child. As such, you should be careful when writing that there was no significant difference in Pb levels between male and female infants - you did not measure the levels in infants but in breast milk. So the trend was that Pb in breast milk was higher in women with male infants than in women with female infants.
Moreover, I am not sure you can extrapolate the sex difference of child's or adults' (adolescents) levels to this comparison in breast milk. This difference in breast milk could be related to the different expression of transporter proteins etc. in mother based on the sex of the fetus (example for arsenic in placenta https://ehjournal.biomedcentral.com/articles/10.1186/s12940-017-0267-8)
but this does not necessarily reflect the difference of male versus female physiology resulting in sex differences in Pb levels of child/adolescent or adult. Contrary, you should find some literature to support the statements you gave (references such as Lee and Ahn just state the gender difference in adolescents or child which does not support the difference in breast milk).
